# An accurate and efficient measure of welfare tradeoff ratios

**Wenhao Qi***, **Edward Vul, Lindsey J. Powell**

Department of Psychology, University of California San Diego, La Jolla, CA, United States of America

* wqi@ucsd.edu

**Data availability statement:** All data and analysis code in the experiments can be found at https://doi.org/10.5281/zenodo.14563524.

**Funding:** The author(s) received no specific funding for this work.

## Abstract

People's decisions are affected by their interest in others' welfare. They can be motivated both to help and to harm others. The direction and magnitude of these motivations can be quantified relative to a person's self-interest as a welfare tradeoff ratio (WTR). This construct is valuable for testing quantitative theories of social motivation. However, most existing measures of WTRs, and the similar construct of social value orientation (SVO), are based on multiple choices between discrete sets of payoffs, which forces a tradeoff between the accuracy and efficiency of the measures. Here we introduce the Lambda Slider, a WTR measure that is simultaneously accurate and efficient. A participant uses a linear slider to choose from a continuous range of payoff allocations for herself and her social partner. The underlying payoff functions for self and other create a one-to-one correspondence between the participant's potential WTR values and the slider positions that she could choose, which enables accurate measurements of WTR from a single response. Across three experiments, we show that a single response on the Lambda Slider has high reliability, high convergent validity with other measures of social motivation, and high external validity for an altruistic decision with real-world consequences. The Lambda Slider is easy to implement and can be applied in a wide variety of studies on the forces that shape social motivation.

## Introduction

People's lives are full of choices that affect both their own welfare and others' welfare. For example, the decision to give your coat to another person on a cold winter night decreases your own welfare but increases that person's welfare. People's decisions in such interdependent situations are driven by a variety of social motivations, including an interest in social norms and reputation along with direct concern for others' well-being [1,2].

The sum of these social forces results in an overall motivation to increase or decrease another person's welfare; i.e., to benefit or harm that person. The direction and magnitude of this motivation can be captured as a welfare tradeoff ratio (WTR): the amount of personal welfare one is willing to give up in order to increase or decrease another person's welfare by a specified amount [3]. Formally, if Alice has a relationship with Bob, then we can express Alice's utility for a given decision as

$$u = w_s + \lambda w_t, \tag{1}$$

**Competing interests:** The authors have declared that no competing interests exist.

where $w_s$ ("s" stands for "self") is Alice's resulting welfare (her actual or expected payoff from the decision), $w_t$ ("t" stands for "target"; we use "target" instead of the usual "other" due to the confusability between the letter o and the number 0 as subscripts, and that the letter t happens to be the alphabetical successor of the letter s) is Bob's welfare (his actual or expected payoff), and $\lambda$ is Alice's welfare tradeoff ratio toward Bob. For conciseness, we will use $\lambda$ to represent welfare tradeoff ratios throughout the paper. A higher $\lambda$ indicates stronger altruism or friendliness on the part of Alice toward Bob, as it means that Alice will favor actions or situations that are good for Bob, even at the expense of some of her own welfare. In contrast, a lower $\lambda$ indicates stronger selfishness or dislike. A negative $\lambda$ would mean that Alice could perceive utility in sacrificing some of her own welfare in order to harm Bob.

An important goal for social psychology and behavioral economics has been to understand the factors that impact people's concern for others' welfare [4–8]. WTRs are one of several dependent variables researchers have developed for measuring this concern. Others include decisions in specific economic games (e.g., the dictator game) and composite constructs such as "social value orientation" (SVO). The continuous version of the SVO construct is similar, formally and conceptually, to $\lambda$ [7]. One advantage of both $\lambda$ and SVO is that they measure generalizable values that can be used to predict people's choices across many decision contexts.

An ideal tool for measuring social motivation would be both accurate and efficient. (For simplicity, in most of this paper, "accuracy", "accurate" or "accurately" entails the technical concepts of both accuracy (or unbiasedness) and precision (or reliability); i.e., a measure needs to be both accurate and precise in order to be called "accurate". For our purposes, higher efficiency means fewer responses from a participant for one measurement.) This would make it feasible for researchers to measure meaningful differences or changes in concern for others across many people, partners, or situations. For example, an accurate and efficient measure of $\lambda$ could allow researchers to study how $\lambda$ changes as social partners build a history of reciprocation by quickly and repeatedly sampling across interactions [9]. Or it could allow researchers to study how $\lambda$ reflects positions and connections among many people in a social network [10].

Existing measures of both $\lambda$ and SVO have generally faced a tradeoff between accuracy and efficiency. As described below, they achieve accurate estimates of one participant's concern for another person by asking the participant to make a high number of decisions about how they would allocate payoffs with that person. Here we propose a new measure of $\lambda$, the Lambda Slider, that largely avoids this tradeoff and achieves an accurate measurement of $\lambda$ from only a single decision. We first explain why the accuracy–efficiency tradeoff arises in existing measures of $\lambda$ and SVO and how our new measure eliminates this tradeoff. We then present three experiments testing the psychometric properties of the Lambda Slider.

## Binary allocation tasks

How can we measure $\lambda$? The simplest way is through a "binary allocation task" [11]. If we want to measure Alice's $\lambda$ toward Bob, we can give Alice two allocation options to choose from (Fig 1A). Option A results in $5 for Alice and $0 for Bob ($w_s = 5$, $w_t = 0$), while Option B results in $0 for Alice and $10 for Bob ($w_s = 0$, $w_t = 10$). (Here we assume a linear relationship between monetary payoffs and welfare, and that the same increase in payoff leads to the same increase in welfare for both oneself and the target. In practice this may not be exactly true [12], but in our experiments we try to minimize sources of nonlinearity.) If $\lambda > 0.5$, Alice will choose Option B since it leads to a higher overall utility for herself than Option A. If $\lambda < 0.5$, then the overall utility of Option A is higher, and Alice will be more likely to choose that

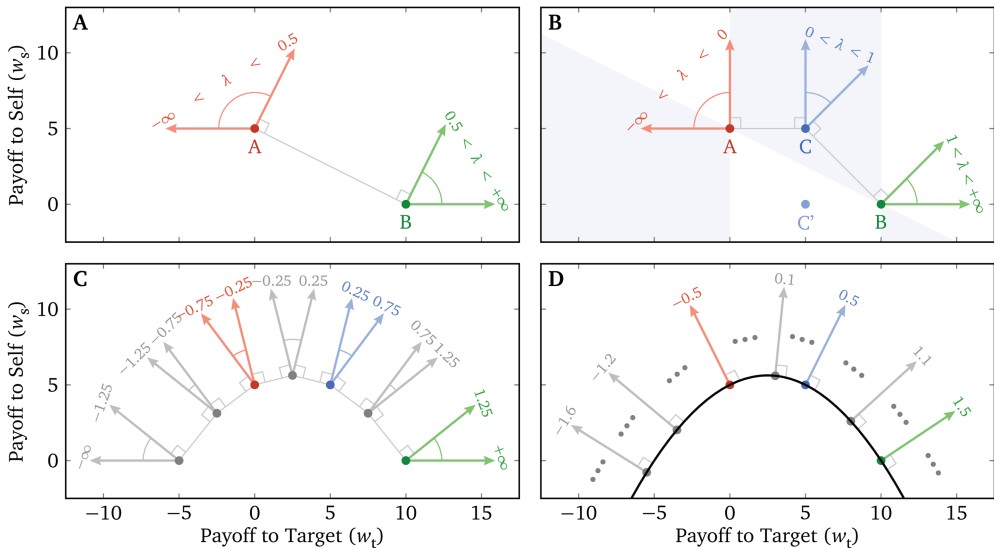

**Fig 1. From binary allocation tasks to the Lambda Slider.** (**A**) A binary allocation task, where the threshold of $\lambda$ for switching between Options A and B is $\hat{\lambda} = 0.5$. The arrows point in the direction of the gradient of the utility function (Eq (1)) for a given $\lambda$. (**B**) Adding a third option C to create a triple-dominance task, which is equivalent to two binary allocation tasks with $\hat{\lambda} \in \{0, 1\}$, but requires only one response. The shaded areas are all the locations where Option C can be placed in order for the task to be triple-dominance. C′ is a hypothetical third option that does not form a triple-dominance task with A and B because they do not fall along a strictly concave function $w_s = f(w_t)$. (**C**) An illustration of a hypothetical "septuple-dominance task" in which each option would be preferred for some range of $\lambda$. (**D**) A possible option space for a Lambda Slider, where a participant can choose any point on the curve (via a slider; Fig 2A). Each point on the curve corresponds to a unique $\lambda$ whose corresponding utility gradient is perpendicular to the tangent of the curve at that point.

option instead. Therefore, Alice's decision on this allocation task tells us whether her $\lambda$ toward Bob is above or below the threshold $\hat{\lambda} = 0.5$. This threshold tested by the task can be adjusted by changing the payoff values involved in the two allocation options.

However, a single binary allocation task has very low sensitivity, defined as the inverse of the smallest change that can be detected by the measure. A binary allocation task with $\hat{\lambda} = 0.5$ cannot distinguish among different $\lambda$s above, or below, 0.5. By analogy, refusing to give your coat to another person could reflect any $\lambda$ ranging from valuing your own warmth just a little more than theirs to actively wishing for them to be cold. The accuracy of a measure is upper-bounded by its sensitivity. To gain a higher sensitivity in our overall measurement of $\lambda$, we can give Alice *multiple* binary allocation tasks with different $\hat{\lambda}$s. For instance, if we assume $\lambda$ falls between –2 and 3 and want a measure that gets within 0.5 of the correct value, we need 9 tasks with $\hat{\lambda} \in \{-1.5, -1, \dots, 2.5\}$. If we aim to get within 0.1 of the correct value, then the number of tasks goes up to 49. This illustrates the inevitable tradeoff between sensitivity and efficiency when we measure $\lambda$ with binary allocation tasks.

Most existing measures of $\lambda$ [13–15], or related constructs such as social value orientation (SVO; [6,11,16–19]), share the logic of narrowing down $\lambda$ with multiple binary allocation tasks, thus sharing the tradeoff between sensitivity and efficiency. (We believe that fundamentally $\lambda$ and SVO are the same thing. Their difference is mostly historical—SVO was traditionally treated as categorical and describes a person's stable disposition toward a unidentified other, while $\lambda$ is usually treated as continuous and specific to each decision influenced by a variety of factors. Recent work has shown a convergence between these two concepts (e.g.,

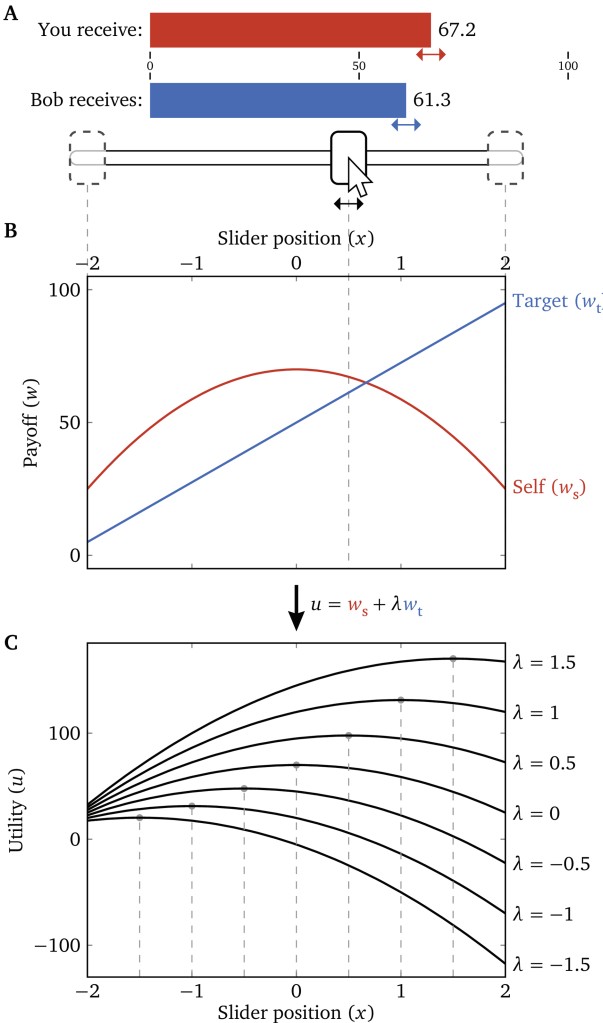

**Fig 2. The (quadratic) Lambda Slider.** (**A**) The interface of the slider. The payoff to oneself (red bar) and payoff to the target (blue bar) change continuously as the participant moves the slider. (**B**) The payoff functions of the quadratic Lambda Slider used in Experiment 1 ($a = 11.25$, $b_s = 70$, $b_t = 50$, $x_{min} = -2$ and $x_{max} = 2$). If we plot $w_s$ and $w_t$ against each other, we get a parabolic curve similar to Fig 1D. (**C**) Examples of the participant's utility function for different $\lambda$s. The slider position that maximizes each utility function is marked, which is an identity function of the participant's $\lambda$.

[7,9]). For a comprehensive review of the measures in the SVO literature, see [7].) This can result in study designs in which many participants must be recruited to study the effects of only a few factors on social motivation (e.g., [20]).

## Triple-dominance tasks

Can we achieve the level of sensitivity of many binary allocation tasks with only a few responses or even one response from the participant? We can draw inspiration from the Triple-Dominance Measure [6]. Although a triple-dominance task is equivalent to *two* binary allocation tasks, a participant only needs to make *one* response on the measure. We can create a triple-dominance task by adding a third option to the binary allocation task in Fig 1A:

Option C results in $5 for Alice and $5 for Bob (Fig 1B). With these three options, Alice will choose A if $\lambda < 0$, B if $\lambda > 1$, and C if $\lambda$ is between 0 and 1. Therefore, this triple-dominance task is equivalent to *two* binary allocation tasks but Alice only needs to make *one* decision by choosing the best option among the three.

Allocation options in a triple-dominance task must be selected such that for any given $\lambda$, one option dominates (i.e., results in a higher utility than) the other two, and for each option there exists some $\lambda$ such that the given option is dominant. In order to maintain these features, the three options need to fall along a *strictly concave* function $w_s = f(w_t)$. This means that Options A and B constrain the possible payoffs offered in Option C, as illustrated by the shaded areas in Fig 1B. As a counterexample, consider Option C′ in Fig 1B, which corresponds to $w_s = 0$ and $w_t = 5$. Given Options A, B and C′, Alice will choose A if $\lambda < 0.5$ and B if $\lambda > 0.5$, but she will never choose C′, so the task does not satisfy the criterion that for each option there exists some $\lambda$ such that the given option is dominant.

## Lambda Slider

By the same logic, we can add more allocation options to a single-choice task to gain a higher sensitivity in the measurement of $\lambda$. Fig 1C is a hypothetical example of a "septuple-dominance task" with 7 options, and the sensitivity of the measurement is $\frac{1}{0.5}$ for $\lambda \in (-1.25, 1.25)$. The options still need to fall along a strictly concave function $w_s = f(w_t)$ to ensure that each option corresponds to the best choice given some $\lambda$.

If we keep adding options to the task, we can create a smooth, continuous curve in the $w_s$–$w_t$ space (Fig 1D), with each point on the curve corresponding to a *single* $\lambda$. This one-to-one correspondence (bijection) between potential $\lambda$s and points on the curve results in a (theoretically) infinite sensitivity of the measurement, which makes it possible to accurately measure a participant's $\lambda$ toward a particular social partner from a single choice. Another way to understand this is to notice that each point on the curve corresponds to one particular exchange rate between $w_s$ and $w_t$, and the participant can vary the exchange rate continuously until she finds the preferred one according to her $\lambda$.

We can present such a continuous set of allocations to the participant with a slider (Fig 2A), and we call it the Lambda Slider (see S1 Appendix for a formal definition and S2 Appendix for comparison with a related measure, the Circle Test [19]). The rewards allocated to the participant and target, $w_s$ and $w_t$, are both continuous functions of the slider position $x$, and we call $w_s(x)$ and $w_t(x)$ the payoff functions of the slider.

We can choose the payoff functions such that the slider position $x$ that a utility-maximizing participant chooses (denoted $x^*$) is an *identity function* of her $\lambda$ toward the social partner in question. Consequently, the slider position is a direct measure of $\lambda$ and no additional calculation is required. One class of such payoff functions (and arguably the simplest class; see S1 Appendix) is

$$w_s(x) = -ax^2 + b_s, \tag{2}$$

$$w_t(x) = 2ax + b_t, \tag{3}$$

$$x \in \left[ x_{\min}, x_{\max} \right], \tag{4}$$

where $a > 0$ is an arbitrary scale parameter that expands or shrinks the range of payoff values, $b_s$ and $b_t$ are arbitrary shift parameters that can offset the participant and target's payoff ranges from one another, and $x_{\min}$ and $x_{\max}$ are boundaries of the slider (Fig 2B). When we

apply the utility definition of Eq (1), we get

$$u(x) = w_s(x) + \lambda w_t(x) \tag{5}$$
$$= -ax^2 + b_s + \lambda(2ax + b_t) \tag{6}$$
$$= -a(x - \lambda)^2 + a\lambda^2 + b_s + b_t\lambda, \tag{7}$$

which is a concave parabola with a peak at $x = \lambda$ (Fig 2C), so it satisfies the criterion

$$x^* = \underset{x \in [x_{\min}, x_{\max}]}{\arg\max}\ u(x) = \lambda, \quad \forall \lambda \in (x_{\min}, x_{\max}). \tag{8}$$

In other words, the participant will choose the slider position that is equal to her $\lambda$ (as long as it falls between $x_{\min}$ and $x_{\max}$) in order to maximize her utility, and this single response on the Lambda Slider gives a measurement of the participant's $\lambda$ with theoretically infinite sensitivity, though of course there will be some limits imposed by the implementation of the task.

We call a Lambda Slider with payoff functions given by Eqs (2) and (3) the quadratic Lambda Slider. When plotted on the $w_s$–$w_t$ plane, the quadratic Lambda Slider is still a parabola, and $w_s = f(w_t)$ is a strictly concave function, similar to Fig 1D. The (quadratic) Lambda Slider shares the logic with mechanism design [21], i.e., we design the payoff structure such that the player's rational action directly reveals her hidden preferences ($\lambda$ in our case). In all the experiments below, the slider position $x$ on the Lambda Slider directly maps to $\lambda$.

## SVO Slider Measure

One apparent difference between the Lambda Slider and the measures based on binary allocation tasks is that the set of possible responses is continuous for the former but discrete for the latter. There is a measure, the SVO Slider Measure, that employs continuous sliders to assess SVO or $\lambda$ [7,18]. This measure consists of 6 sliders, each involving linear payoff functions for the participant, $w_s$, and another person, $w_t$. Each slider connects two points on a circular arc, centered on $(w_s, w_t) = (50, 50)$ (Fig 3A). The points represent the choices most aligned with four categorical social value orientations: competitive, selfish, prosocial, and altruistic. After calculating the average chosen payoffs for self and target across the 6 sliders ($\overline{w_s}$ and $\overline{w_t}$), a summary output is calculated as:

$$\text{SVO}^\circ = \arctan\left(\frac{\overline{w_t} - 50}{\overline{w_s} - 50}\right). \tag{9}$$

This "angle" can be interpreted as the angle of the point a participant would choose among payoff values aligned along the arc in Fig 3A, with larger angles corresponding to higher values of $\lambda$. (In fact, this arc can be used to create a "circular" Lambda Slider; see S2 Appendix.)

The SVO Slider Measure is relatively efficient, requiring 6 responses for one measurement, which is fewer than previous measures such as the 9-item Triple-Dominance Measure [6] and the Ring Measure [16,17]. However, the linear nature of the slider payoff functions effectively results in binary allocation tasks, which create the familiar tradeoff between sensitivity and

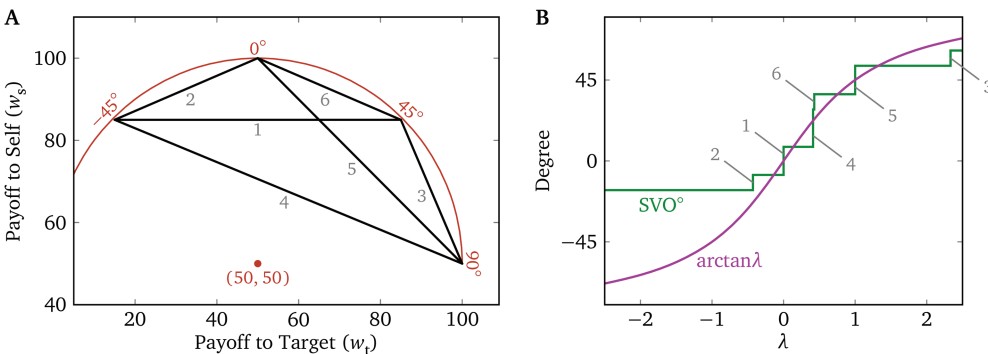

**Fig 3. The SVO Slider Measure** [18]. (**A**) The payoff functions of the 6 primary items of the measure (black lines). Each segment represents the linear relationship between $w_s$ and $w_t$ on one of the items, and they are labeled in the same order as in [18]. The red arc and point $(50, 50)$ provide an intuitive explanation (but not formal justification) for the calculation of SVO°. (**B**) The theoretical step function (green curve) between the output of the measure (SVO°) and $\lambda$. The labeled vertical segments correspond to the $\hat{\lambda}$s (the thresholds of $\lambda$ at which a utility-maximizing participant switches from one end to the other on the sliders) of the 6 items. The theoretical response on the circular Lambda Slider (arctan $\lambda$; Eq (15) in S2 Appendix) is also plotted for comparison.

efficiency. For instance, the first slider has payoff functions

$$w_s(x) = 85, \tag{10}$$
$$w_t(x) = -70x + 85, \tag{11}$$

where $x \in [0, 1]$ is the slider position. Then the utility function is

$$u(x) = w_s(x) + \lambda w_t(x) \tag{12}$$
$$= -70\lambda x + 85(1 + \lambda). \tag{13}$$

A utility-maximizing participant would choose $x = 0$ if $\lambda > 0$, choose $x = 1$ if $\lambda < 0$, and be indifferent if $\lambda = 0$. Therefore, this slider is equivalent to a binary allocation task with $\hat{\lambda} = 0$. Similarly, the $\hat{\lambda}$s for the remaining 5 sliders are $-\frac{3}{7}$, $\frac{7}{17}$, $\frac{3}{7}$, $1$, and $\frac{7}{3}$. The measure has no way of distinguishing among different $\lambda$s between two adjacent $\hat{\lambda}$s (e.g., 0 from slider 1 and $\frac{7}{17}$ from slider 4). For any given $\lambda$, we can derive the output of the measure, SVO°, from the choices that the participant would make on the 6 sliders, which is plotted in Fig 3B. The relationship between $\lambda$ and SVO° is not one-to-one, but many-to-many (i.e., different $\lambda$s between two adjacent $\hat{\lambda}$s lead to the same responses, and for a given $\lambda$ that is equal to one of the $\hat{\lambda}$s, all positions on one of the sliders are equally preferable). Technically speaking, the SVO Slider Measure has high resolution but low sensitivity. The Lambda Slider has the potential to provide both higher sensitivity and higher efficiency, though when implemented as a single item measure it may have somewhat lower reliability.

## Current research

In Experiment 1, we compare the Lambda Slider to the SVO Slider Measure in terms of test–retest reliability and convergent validity, because (a) the SVO Slider Measure performs relatively well in practice and is regarded as the state-of-the-art measure of $\lambda$, and (b) it can share an interface with the Lambda Slider (Fig 2A), allowing us to easily mix them in a single experiment. (Future work can compare the Lambda Slider with other popular measures of $\lambda$, such

as the Welfare Trade-Off Task [14,15], which requires a different interface.) In Experiment 2, we rule out an alternative hypothesis that participants use a heuristic to make decisions on the Lambda Slider. In Experiment 3, we test the external validity of the Lambda Slider using a social decision with real-world consequences, and explore the effects of inequity aversion on measurements of $\lambda$. All data and analysis code in the experiments can be found at https://doi.org/10.5281/zenodo.14563524, with instructions for reproducing the results.

## Experiment 1

We have formally shown above that the one-shot Lambda Slider has infinite sensitivity. However, how much such theoretical *sensitivity* translates to empirical *accuracy* is limited by the degree to which participants perfectly maximize a utility function in the form of Eq (1).

In Experiment 1, we evaluate the reliability and validity of the quadratic Lambda Slider, and compare it with the SVO Slider Measure. (In all three experiments, we report all measures, manipulations and exclusions.) To evaluate the psychometric properties of the Lambda Slider, we need to elicit as wide a range of $\lambda$s as possible from each participant. It has been shown that a person's $\lambda$ toward another person decreases as their social distance increases [13]. Therefore, we asked participants to each generate a list of 10 known people (subsequently called "targets") occupying a range of social distances from themselves. We then had participants make hypothetical allocation decisions between themselves and each of those 10 targets. Such a manipulation not only helps elicit a wide range of $\lambda$s, but also tests the measure's convergent validity with social distance, based on an expected negative correlation between a participant's measured $\lambda$s toward the targets and her reported social distances from the targets.

### Methods

**Participants.** 40 participants were recruited on Prolific and completed the experiment online between May 7 and 10, 2022. (The sample sizes in all experiments were determined before any data analysis, although this is not strictly necessary because all data analyses are fully Bayesian. The sample sizes of Experiments 1 and 2 were determined heuristically, while the sample size of Experiment 3 was determined based on a frequentist power analysis as preregistered.) The participants were drawn from the "standard sample", were located in the USA, were fluent in English, had an approval rate of at least 95%, and had at least 10 previous submissions on the platform. The participants gave informed consent to participate in the experiment by clicking a button on the web page displaying the consent form at the start of the experiment. The experiment was approved by the UCSD institutional review board (Protocol #800709). Each participant received US$2 for completing the experiment. 30 participants (7 female, 23 male) passed at least 8 out of the 9 attention checks (see below) and only these participants are included in the analyses below.

**Design.** The experiment is implemented as a web page and can be viewed at https://experiments.evullab.org/qi-games-2/. There are three stages in the experiment: List, Rank, and Slide.

In the List stage, participants are asked to list the first names of 10 people they know, 2 in each of 5 categories: family+, friends, neighbors and colleagues, acquaintances, and adversaries. These categories are designed to maximize the range of social distances between a participant and the targets and, presumably, of the participant's $\lambda$s toward the targets.

In the Rank stage, participants are asked to rank the 10 names they input in the List stage "based on how close you are to them (in terms of relationship, not physical distance)" by

dragging the 10 names in a vertical list. The order of the names is initially randomized. The final order of the names is recorded.

In the Slide stage, each participant completes 72 allocation trials using an interface similar to Fig 2A. In each trial, participants drag the horizontal slider, and the payoffs to the participant ($w_s$) and to the target ($w_t$), depicted both numerically and as horizontal bars, change continuously according to the underlying payoff functions, which are bounded at 0 and 100 in an arbitrary unit. The bars are labeled "You receive:" and "[Target] receives:", where "[Target]" is replaced by the name of the target in the current trial. Participants are told that the payoffs are hypothetical and are asked to move the slider until the settings look the best to them. The initial position of the slider is randomized in each trial.

In order to evaluate the test–retest reliability of the Lambda Slider and the SVO Slider Measure, we need two measurements for each target for each measure, which amounts to 2 quadratic Lambda Slider trials and 12 SVO Slider Measure trials (twice for each of the 6 primary items) per target. If we measured each participant's $\lambda$s toward all the targets on both measures, there would be 6 times as many SVO Slider Measure trials as Lambda Slider trials and too many trials in total. Therefore, we measure each participant's $\lambda$s toward all the 10 targets on the Lambda Slider (20 trials in total), but only targets whose social distance rankings are 1, 4, 7 or 10 on the SVO Slider Measure (48 trials in total).

A participant's response on each quadratic Lambda Slider trial is directly used as the measured $\lambda$ by virtue of Eq (8). A participant's responses on the 6 different SVO Slider Measure items are aggregated to an SVO° according to Eq (9). The first occurrence of each item is treated as part of the first measurement of SVO°, and the remaining items compose the second measurement.

We also include 4 "catch trials" as attention checks, in which "[Target]" is replaced by "Left" or "Right". Participants are instructed that on these trials they should move the slider to the far left (right) regardless of the payoffs. A participant is considered to pass a catch trial if the slider position she chooses satisfies $x < -1.9$ ($x > 1.9$) when the target is "Left" ("Right"). These 72 trials are randomized in order. Immediately after Trials 2, 6, 14, 30 and 62 (called "memory trials"), participants are asked to type the target's name (or "Left" or "Right") they just saw as attention checks. Participants are considered to pass a memory trial if the name they type is the same as the target's name they just saw, after transforming both names to lowercase and removing whitespaces. The combined "catch" and "memory" trials result in 9 attention checks altogether.

The quadratic Lambda Slider trials have payoff functions defined by Eqs (2)–(4) with $a = 11.25$, $b_s = 70$, $b_t = 50$, $x_{min} = -2$ and $x_{max} = 2$, such that $w_s \in [25, 70]$, $w_t \in [5, 95]$, and the range of $\lambda$ that can be accurately measured is $(-2, 2)$ (Fig 2B). We make the range of $w_s$ and $w_t$ narrower than the full range $[0, 100]$ because (a) allowing the payoffs to reach extreme values creates salient points that may bias participants' responses [22], and (b) the welfare participants perceive for themselves and the targets with respect to the raw payoffs is likely to be more nonlinear when the payoffs are close to 0 [12]. The catch trials depict payoff functions in the same manner as the Lambda Slider trials. The SVO Slider Measure trials have the same payoff functions as in [18], as shown in Fig 3A.

## Results

**Test–retest reliability.** We evaluate the test–retest reliability of the quadratic Lambda Slider by estimating the correlation between the two measurements of $\lambda$ of each participant–target combination, and compare it to the correlation between the two measurements of SVO°. We do not expect the correlation of $\lambda$s to be as high as the correlation of SVO°s

because (a) one measurement of SVO° is an aggregation of 6 responses, which almost certainly has less noise than 1 response on the Lambda Slider, and (b) the Lambda Slider has a nonlinear payoff structure, which might be harder to understand than the linear payoff structures of the SVO Slider Measure. However, researchers using the Lambda Slider have the flexibility to select the number of repeated measurements to achieve the desired tradeoff between precision and efficiency. (This is different from the tradeoff between sensitivity and efficiency involved in the binary allocation tasks, mentioned in the Introduction. For the binary allocation tasks, the tradeoff arises from a theoretical limitation which even applies to a noiseless decision maker, while the current tradeoff is only due to noise in the decisions.) We will first compare the test–retest reliability of the 1-response $\lambda$ with the 6-response SVO°, and then estimate the reliability of the multiple-response $\lambda$.

Figs 4A and B plot the relationship between the two measurements of each participant–target combination. We fit a bivariate normal distribution to the Lambda Slider data and to the SVO Slider Measure data (see S4 Appendix for details; all data analyses are fully Bayesian, and we use uninformative or weakly informative priors based on null hypotheses for the main parameters). The mean and standard deviation of one measurement are $\mu$ = 0.19 and $\sigma$ = 1.11 for the Lambda Slider data and $\mu$ = 16.5 and $\sigma$ = 25.4 for the SVO Slider Measure data. (When the uncertainty in the estimation of a parameter is not important, like $\mu$ or $\sigma$ here, we report a single number which is the posterior median of the parameter. Otherwise, see below.) The two measurements on the quadratic Lambda Slider have a high correlation ($\rho$ = 0.859 (0.823, 0.888), Fig 4A), indicating that a single measurement on the Lambda Slider has high test–retest reliability. (Here, 0.859 is the posterior median of $\rho$, and (0.823, 0.888) is the 95% (equal-tailed) credible interval of $\rho$. The same notation is used for the rest of the paper. Besides, we do not report the probability of direction ($p_d$) or the Bayes factor (relative to a null model) of a parameter if $p_d$ calculated using the "direct" method [23] is 100%, in which case the true $p_d$ is expected to be at least 99.975% because we take at least 4000 posterior samples in our models.) As predicted, the two measurements of SVO° have an even higher correlation ($\rho$ = 0.952 (0.930, 0.967), Fig 4B).

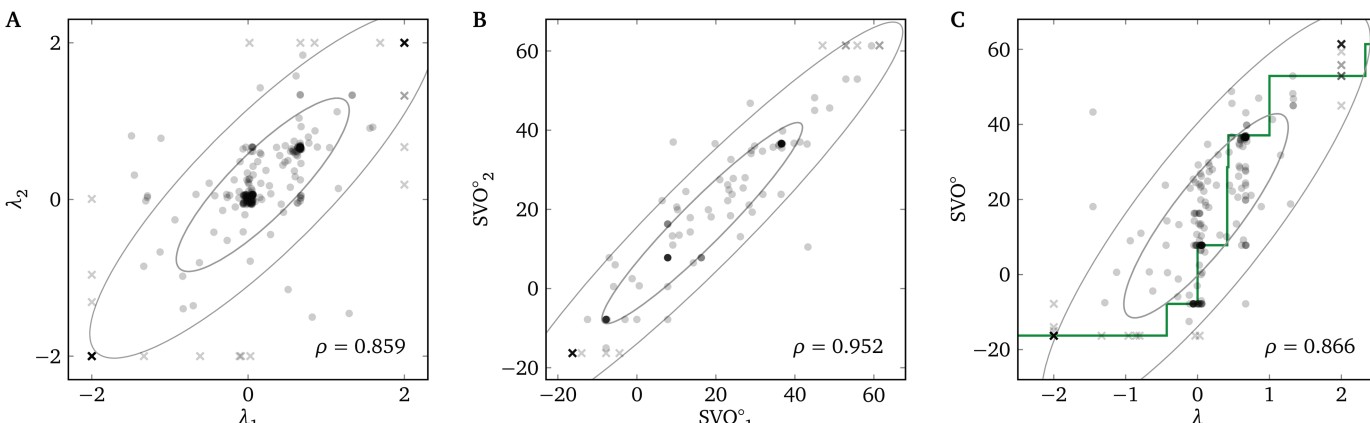

**Fig 4. Test–retest reliability of (A) the quadratic Lambda Slider and (B) the SVO Slider Measure, and (C) convergent validity between the two measures.** In (**A**) and (**B**), each data point represents one participant–target combination. In (**C**), for each participant–target combination, there are two data points representing the first $\lambda$ paired with the first SVO°, and the second $\lambda$ paired with the second SVO°. The green line is the theoretical relationship between $\lambda$ and SVO°, same as Fig 3B. Data points on the boundaries, which are treated as censored data, are represented as crosses (same for all figures below). The ellipses indicate the 1-$\sigma$ and 2–$\sigma$ iso-density loci of the fitted bivariate normal distributions with parameters set to their posterior medians.

To assess how many administrations of the Lambda Slider would be required to make the reliability scores of the two measures comparable, we estimate the reliability score of the average of multiple measurements on the Lambda Slider. According to the classical test theory [24], the test–retest correlation is equal to the reliability score, and

$$\rho = \frac{\sigma_t^2}{\sigma_t^2 + \sigma_e^2}, \tag{14}$$

where $\sigma_t^2$ is the variance of the true score and $\sigma_e^2$ is the variance of the error (of one measurement) on the Lambda Slider. Let $\rho'$ be the reliability score of the average of $n$ measurements on the Lambda Slider. Averaging $n$ measurements shrinks the variance of the error by a factor of $n$, so we have

$$\rho' = \frac{\sigma_t^2}{\sigma_t^2 + \frac{\sigma_e^2}{n}}, \tag{15}$$

and therefore

$$\frac{\rho'}{1 - \rho'} = n \frac{\rho}{1 - \rho}. \tag{16}$$

The same result can be obtained by first assuming a multivariate normal distribution over $2n$ variables with the same mean and standard deviation and a fixed pairwise correlation $\rho$, and then deriving the correlation between the mean of the first $n$ variables and the mean of the other variables.

For a baseline of $\rho = 0.858\ (0.823, 0.888)$, if we increase the number of measurements to $n = 3$, we have $\rho' = 0.948\ (0.933, 0.960)$, which indicates that the reliability of the average of 3 measurements on the Lambda Slider is expected to be comparable to the reliability of the SVO Slider Measure, which requires 6 measurements.

**Convergent validity: Lambda Slider vs. SVO Slider Measure.** Fig 4C plots the relationship between $\lambda$ as measured by the quadratic Lambda Slider and by the SVO Slider Measure (SVO°). We fit a 4-variate normal distribution (2 measurements × 2 measures for each participant–target combination) to the data (see S4 Appendix for details). The two measures are highly correlated ($\rho_{\lambda\nu} = 0.866\ (0.821, 0.902)$), indicating that the Lambda Slider has high convergent validity with the SVO Slider Measure.

In Fig 4C, there seem to be more responses of SVO° between 7.82° and 36.61° than $\lambda$ between 0 and 0.667. This does not indicate that the SVO Slider Measure has a higher sensitivity for measuring $\lambda$ in this range than the Lambda Slider, because (a) it is inconsistent with theoretical predictions, and (b) it can be explained away by assuming that a participant probabilistically chooses between self-gain maximization and perfect inequity aversion for *each decision*, which we do not explicate here but can be investigated by future work.

**Convergent validity: $\lambda$ vs. social distance.** Fig 5 shows participants' measured $\lambda$s from the Lambda Slider and their SVO° measurements toward targets with different social distances from the participants. We fit a Bayesian mixed effects model to the data with the social distance ranking as a monotonic predictor and $\lambda$ or SVO° as the dependent variable (see S4 Appendix for details). As predicted, $\lambda$ as measured by the quadratic Lambda Slider decreases as the target's social distance ranking increases (mean slope $b_3 = -0.25\ (-0.34, -0.16)$; this corresponds to how much $\lambda$ decreases on average as social distance ranking increases by 1). The output of the SVO Slider Measure (SVO°) also decreases as the target's social distance ranking increases (mean slope $b_3 = -16.9\ (-20.9, -13.0)$; this corresponds to how much SVO° decreases on average as social distance ranking increases by 3).

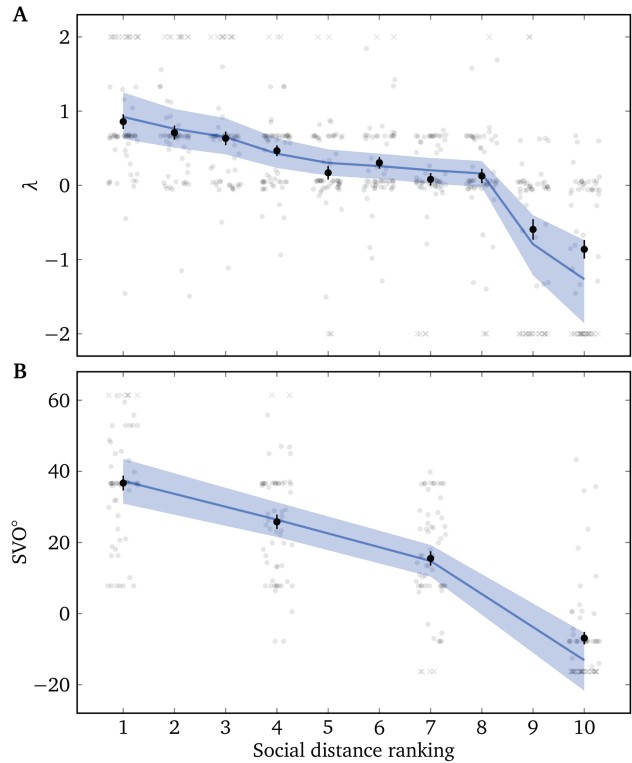

**Fig 5. Relationship between $\lambda$ and social distance, for the quadratic Lambda Slider (A) and the SVO Slider Measure (B).** Each raw data point is one of the two measurements of a participant–target combination. Black points and ranges represent the means and standard errors of data in each group. Blue lines and ranges represent the conditional effects (also called marginal effects; [25]) of the social distance ranking as a monotonic predictor, with 95% credible intervals.

It is worth noting that participants' $\lambda$s spanned a wide range (Fig 5A). The *mean $\lambda$* toward the socially closest person is 0.86, which means that participants value the target's welfare almost as much as their own. The *mean $\lambda$* toward the socially most distant person is –0.86, which means that participants are almost willing to give up \$1 to take \$1 away from the target. The SVO Slider Measure has very low sensitivity for $\lambda < -0.43$ or $\lambda > 1$ (Fig 3B), and thus cannot measure a large subset of plausible $\lambda$s accurately.

## Experiment 2

Experiment 1 provided evidence that the Lambda Slider is a valid and reliable measure of $\lambda$. However, it is possible that instead of making decisions by incorporating the relevant $\lambda$s into a utility function like the one in Eq (1) (we call this hypothesis $H_\lambda$), participants use the slider position as a qualitative representation of kindness/spitefulness and make decisions based on this representation (we call this hypothesis $H_\chi$). For instance, after getting an intuitive idea of how the two payoffs change as a function of the raw slider position $\chi \in [0, 1]$, a participant might treat $\chi$ = 0, 0.25, 0.5, 0.75 and 1 as "very mean", "somewhat mean", "neutral", "somewhat nice", and "very nice", respectively. (Note that $\chi$ is different from $x$ above. $\chi = 0$ ($\chi = 1$) always corresponds to the left (right) end of the slider.) Then she may choose to be "very nice" to Alice, "somewhat mean" to Bob, etc., and choose slider positions accordingly.

$H_\lambda$ and $H_\chi$ make different predictions when we alter the relationship between $\lambda$ and the raw slider position $\chi$. For example, suppose that in an initial trial in which $x \in [-2, 2]$,

Alice chooses $\chi = 0.75$ on the quadratic Lambda Slider, corresponding to a $\lambda$ of 1 for that target. If we then have Alice make a decision for the same target on a different quadratic Lambda Slider with $x \in [-1, 1]$, $H_\lambda$ predicts that she will choose $\chi = 1$ (corresponding to $\lambda = 1$) but $H_\chi$ predicts that she will still choose $\chi = 0.75$ (corresponding to $\lambda = 0.5$).

In general, suppose we have two quadratic Lambda Sliders, Slider A and Slider B. Let $x \in [x_{\min A}, x_{\max A}]$ on Slider A and $x \in [x_{\min B}, x_{\max B}]$ on Slider B. Let the raw slider position that the participant chooses be $\chi_A$ on Slider A and $\chi_B$ on Slider B. For simplicity, suppose neither $\chi_A$ nor $\chi_B$ is at the boundaries of the slider. Let $\lambda_A$ ($\lambda_B$) be the $\lambda$ derived from $\chi_A$ ($\chi_B$). We have

$$\lambda_A = (1 - \chi_A)x_{\min A} + \chi_A x_{\max A}, \tag{17}$$

$$\lambda_B = (1 - \chi_B)x_{\min B} + \chi_B x_{\max B}. \tag{18}$$

Given $H_\lambda$, since $\lambda$ on the two sliders should be the same, we have $\lambda_A = \lambda_B$, and therefore

$$\chi_B = \frac{x_{\max B} - x_{\min B}}{x_{\max A} - x_{\min A}}\chi_A + \frac{x_{\min A} - x_{\min B}}{x_{\max A} - x_{\min A}}. \tag{19}$$

Given $H_\chi$, we have

$$\chi_B = \chi_A. \tag{20}$$

To adjudicate between $H_\lambda$ and $H_\chi$, in Experiment 2, we let participants make decisions for each target on three different quadratic Lambda Sliders with different ranges of $x$, and see which hypothesis best predicts the responses.

## Methods

**Participants.** 20 participants were recruited on Prolific and completed the experiment online on June 28 and August 5, 2022. The participant consent, experiment approval, pre-screening, payments, and attention check criteria were the same as Experiment 1. 16 participants (4 female, 12 male) passed at least 8 out of the 9 attention checks and are included in the analyses below.

**Design.** Similar to Experiment 1, Experiment 2 is implemented as a web page and can be viewed at https://experiments.evullab.org/qi-games-4/. It also has three stages: List, Rank and Slide, and the List and Rank stages are identical to Experiment 1.

In the Slide stage, there are three quadratic Lambda Sliders: a "base" slider with $x \in [-2, 2]$, same as Experiment 1; a "**pos**itive-shift" slider with $x \in [-1.25, 2.75]$; a "**neg**ative-shift" slider with $x \in [-2.75, 1.25]$ (Fig 6; how we select these ranges and the payoff functions is detailed in S3 Appendix). For each participant, each target is measured twice on each of the three sliders, with a total of 60 Lambda Slider trials. There are 4 "Left"/"Right" catch trials, similar to Experiment 1, whose payoff functions are the same as the base slider. These 64 trials are randomized in order. The memory trials are at the same locations as in Experiment 1, so there are still 9 attention checks altogether.

We will compare the responses on the three sliders. From Eqs (19) and (20), we see that $H_\lambda$ predicts

$$\chi_{base} = \chi_{pos} + 0.1875 = \chi_{neg} - 0.1875, \tag{21}$$

while $H_\chi$ predicts

$$\chi_{base} = \chi_{pos} = \chi_{neg}. \tag{22}$$

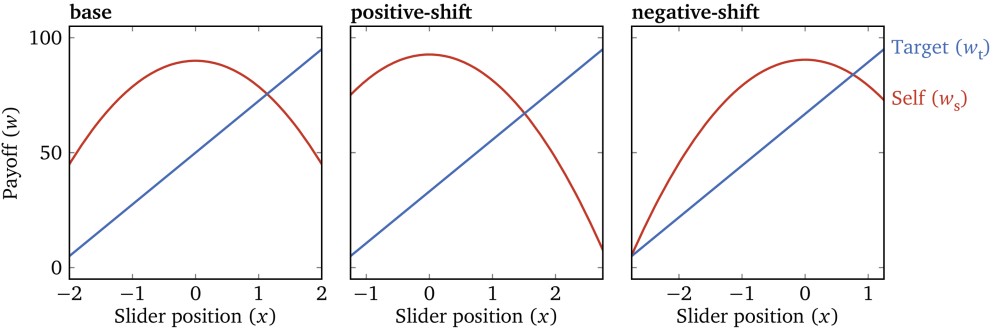

**Fig 6. Payoff functions of the three quadratic Lambda Sliders in Experiment 2.** The ranges on the $x$ axes reflect the ranges of the sliders.

## Results

Fig 7A shows the empirical distributions of raw slider positions on the three different sliders. The means of the three distributions are significantly different and closer to predictions of $H_\lambda$ than $H_\chi$. The deviation from predictions of $H_\lambda$ could be partly due to (a) limited ranges of the sliders (e.g., $\chi_{\text{base}} = 1$ would correspond to $\chi_{\text{neg}} = 1.1875$, which would be clipped to $\chi_{\text{neg}} = 1$), and (b) inequity-averse responses falling halfway between the predictions of $H_\lambda$ and $H_\chi$ (see S3 Appendix). For a more fine-grained analysis, Fig 7B and C plot the comparisons of the responses on the base slider versus the positive-shift or negative-shift slider, and compares them to the predictions of $H_\lambda$ and $H_\chi$. For either hypothesis, we fit a 6-variate normal distribution (2 measurements × 3 sliders for each participant–target combination) to the data, with the constraint that the means satisfy either Eq (21) or Eq (22) depending on the hypothesis (see S4 Appendix for details). The logarithm of the Bayes factor between $H_\lambda$ and $H_\chi$ is 101.6, indicating decisive evidence in favor of $H_\lambda$ compared to $H_\chi$. This

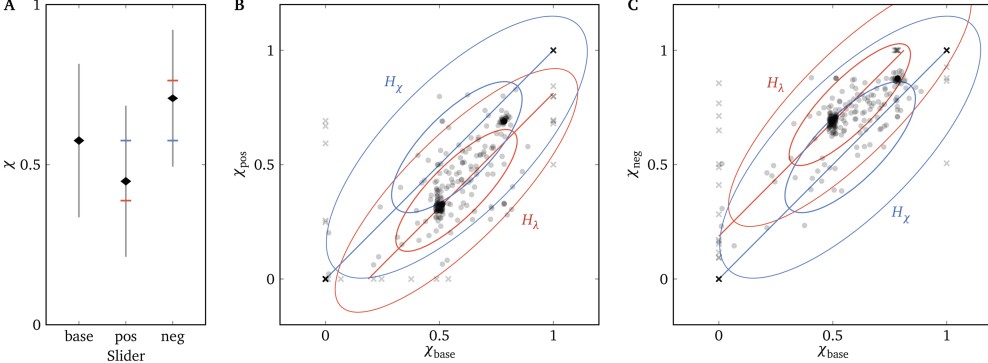

**Fig 7. Results of Experiment 2.** (**A**) Empirical distributions of raw slider positions ($\chi$) for the base, **pos**itive-shift and **neg**ative-shift sliders. The black rhombuses represent means and standard errors, and the gray lines represent standard deviations. For reference, the red and blue bars mark predictions of $H_\lambda$ and $H_\chi$ (Eqs (21) and (22)), respectively, given the mean on the base slider. (**B** and **C**) Responses compared to predictions of $H_\lambda$ and $H_\chi$. For each participant–target combination, there are two raw data points in each panel representing the two measurements on either slider. The diagonal lines indicate the predictions of the two hypotheses without noise. The ellipses indicate the bivariate normal distributions representing the two fitted models (see S4 Appendix).

confirms that participants likely made decisions based on $\lambda$ and utility maximization rather than based on a qualitative representation of kindness.

As further evidence for the test–retest reliability of the Lambda Slider under different configurations, the within-slider correlations under $H_\lambda$ are $\rho_{\text{base}} = 0.876\,(0.833, 0.908)$, $\rho_{\text{pos}} = 0.903\,(0.870, 0.927)$, and $\rho_{\text{neg}} = 0.906\,(0.871, 0.931)$. The between-slider correlations under $H_\lambda$ are $\rho_{\text{bp}} = 0.812\,(0.762, 0.855)$, $\rho_{\text{bn}} = 0.825\,(0.775, 0.865)$, and $\rho_{\text{pn}} = 0.781\,(0.715, 0.832)$, indicating that measurements of $\lambda$ are relatively robust to different range and shift parameters. We also examine the relationship between $\lambda$ and the social distance ranking with a model similar to Experiment 1 (see S4 Appendix), and the mean slope is $b_3 = -0.22\,(-0.34, -0.10)$, $p_d = 99.98\%$.

## Experiment 3

So far, the decisions participants made in the experiments were all hypothetical. However, the utility of the Lambda Slider in practice also depends on its external validity (also called predictive validity by some); i.e., whether hypothetical decisions on the Lambda Slider predict real-world altruistic behavior. Despite theoretical concerns about whether decisions with hypothetical payoffs can predict decisions with real payoffs [12], experiments using matched designs have generally found good alignment between the two settings [26–30]. However, most decisions people make in the lab, such as making monetary tradeoffs in an economic game, are so different from real-life decisions that it is unclear whether behavior in these decisions can generalize to real-life situations. Therefore, for the best test of the external validity of a measure, we need to use real-life decisions with real payoffs. [31] examined the external validity of the SVO Slider Measure using a standard dictator game. Using anonymous targets in all measures, they found that the correlation between the SVO Slider Measure (in terms of SVO°) and amount given in the dictator game was 0.42. Likewise, in Experiment 3, we let participants make a real-life decision of how much money to donate, with an underlying structure of a dictator game, and examine its relationship with hypothetical decisions on the Lambda Slider. We also examine the robustness of the Lambda Slider under different configurations and the effects of inequity aversion on the measurements. This experiment was preregistered at https://osf.io/zbw8f.

### Inequity aversion

A basic assumption of any study on $\lambda$ is that a person's utility function is a *linear* combination of $w_s$ and $w_t$, at least within the range of payoffs in that study. In other words, the only motivations under consideration are the motivations to increase or decrease one's own and the other person's welfare. However, another relevant social motivation is inequity aversion, which is the desire to decrease the absolute difference between $w_s$ and $w_t$ [1]. We can see inequity aversion at play in the previous experiments. In Figs 4A, C and 5A, instead of forming a smooth distribution between $\lambda = 0$ and $\lambda = 2$, many Lambda Slider responses were concentrated at $\lambda = 0.667$, which leads to $w_s = w_t$ given the parameters in Experiment 1. Likewise, in Figs 4B, C and 5B, instead of forming a smooth distribution between SVO° = 7.82° (corresponding to a $\lambda$ slightly greater than 0) and SVO° = 61.39° (the maximum possible value), many SVO Slider Measure responses were concentrated at SVO° = 36.61°, which is consistent with the responses of a perfectly inequity-averse decision maker. Again, in Fig 7, a lot of responses concentrate at the equal-payoff points of the three sliders, which are halfway between the predictions of the two models (see S3 Appendix).

Formally, we can add an inequity-aversion term to the utility function of Eq (1):

$$u = w_s + \lambda w_t - \kappa |w_s - w_t|, \tag{23}$$

where $\kappa \in [0, 1)$ captures the strength of inequity aversion. (It can be shown algebraically that this utility function is equivalent to a utility function with separate advantageous- and disadvantageous-inequality terms but no $\lambda$ term, as in [1]. In fact, the utility function in [1] can always be rewritten in the form of Eq (23), but not vice versa.)

The measurement of $\lambda$ may be biased and may lose sensitivity around the equal-payoff point if the participant has a nonzero $\kappa$. This problem is shared by all the existing measures of $\lambda$, including our Lambda Slider. Trying to counter this problem, [18] describes a set of secondary linear-payoff sliders that are used to distinguish between inequality aversion and "joint gain maximization". Participants' responses to these secondary items can be used to calculate an "inequality aversion index" ranging from 0 (pure inequality aversion) to 1 (pure joint gain maximization). However, this approach can only be used for participants whose responses on the "primary items" (Fig 3A) are consistent with a "prosocial" orientation (i.e., $\lambda \approx 1$), as the index assesses the degree to which a participant's *responses* are closer to a perfectly consistent decision maker with $\lambda = 1, \kappa = 0$ versus $\lambda = 1, \kappa = 1$.

Using a range of payoff configurations, the Lambda Slider can simultaneously measure $\lambda$ and $\kappa$ with no additional restriction on the value of $\lambda$. To see how, we can consider the cases where $w_s \geq w_t$ and $w_s < w_t$ separately and substitute Eqs (2) and (3) into Eq (23):

$$u(x) = \begin{cases} (1-\kappa)w_s + (\lambda+\kappa)w_t & w_s \geq w_t \\ (1+\kappa)w_s + (\lambda-\kappa)w_t & w_s < w_t \end{cases} \tag{24}$$

$$= \begin{cases} (1-\kappa)(-ax^2 + b_s) + (\lambda+\kappa)(2ax + b_t) & w_s \geq w_t \\ (1+\kappa)(-ax^2 + b_s) + (\lambda-\kappa)(2ax + b_t) & w_s < w_t \end{cases} \tag{25}$$

$$= \begin{cases} -a(1-\kappa)\left(x - \dfrac{\lambda+\kappa}{1-\kappa}\right)^2 + \text{const} & w_s \geq w_t \\ -a(1+\kappa)\left(x - \dfrac{\lambda-\kappa}{1+\kappa}\right)^2 + \text{const} & w_s < w_t \end{cases}. \tag{26}$$

For any $\lambda \in \mathbb{R}$ and $\kappa \in [0, 1)$, we can make the difference between the shift parameters of the payoff functions, $b_s - b_t$, positive enough such that $w_s > w_t$ when $x = \frac{\lambda+\kappa}{1-\kappa}$ and thus $x^* = \frac{\lambda+\kappa}{1-\kappa}$; we can also make $b_s - b_t$ negative enough such that $w_s < w_t$ when $x = \frac{\lambda-\kappa}{1+\kappa}$ and thus $x^* = \frac{\lambda-\kappa}{1+\kappa}$ (proof omitted). Assuming that the participant maximizes their utility perfectly, these two different values of $x^*$ allow us to solve for $\lambda$ and $\kappa$ independently. In Experiment 3 we define a likelihood function based on Eq (26) and perform Bayesian inference on $\lambda$ and $\kappa$. Estimating $\kappa$ also allows us to examine *its* external validity, similar to the external validity of $\lambda$, by looking at the relationship between the estimated $\kappa$ and participants' decisions in the dictator game with real payoffs.

## Methods

**Participants.** 90 participants were recruited on Prolific and completed the experiment online on November 20, 2023. The participant consent, experiment approval, prescreening, payments, and attention check criteria were the same as Experiment 1. 76 participants

(39 female, 36 male, 1 unknown) passed at least 4 out of the 5 attention checks and only these participants are included in the analyses below.

**Design.** Similar to Experiments 1 and 2, Experiment 3 is implemented as a web page and can be viewed at https://experiments.evullab.org/qi-games-7/. It has three stages: List, Slide, and Bonus & Donation.

The List stage is the same as Experiment 1, except that participants list only one target in each of the five categories. We use the categories as proxies for the social distance rankings and do not ask the participants to rank the targets. After participants list these targets, we introduce an additional target described as a victim in the wildfires of Maui, Hawaii in 2023. The name of the target was extracted from a non-paywalled news article on the wildfires, and we provide a link to the article as well as a description of the victim's circumstances.

In the Slide stage, there are three quadratic Lambda Sliders: a "balanced" slider, where $w_s$ can be either greater than, less than, or equal to $w_t$ depending on the slider position; a "self-more" slider, where $w_s > w_t$ holds regardless of the slider position (within the allowed range); and a "target-more" slider, where the inverse holds (Fig 8; see S3 Appendix for the exact parameters). The range of $\lambda$ on the sliders is always $[-2, 2]$. These three sliders allow us to estimate the inequity aversion parameter $\kappa$ as described above.

For each participant, a slider allocation to each of the 6 targets is measured twice on each of the 3 sliders, with a total of 36 Lambda Slider trials, which are randomized in order. A "Left" catch trial and a "Right" catch trial (as in Experiment 1) are added, which become Trials 5 and 20, respectively. Trials 2, 11 and 32 are memory trials, so there are 5 attention checks altogether.

In the Bonus & Donation stage, participants are asked to use a slider to split US$2 between a monetary bonus to themselves and a donation to the Maui Strong Fund, a fund created by the Hawaii Community Foundation to support recovery from the Maui wildfires. Participants essentially play a dictator game between themselves and the fund. The slider has a precision of $0.01. Participants are assured that there is no deception involved and that we will actually donate the amount they specify to the Maui Strong Fund. We also tell participants that after we have collected all the data, we will send them a spreadsheet documenting the donation from each participant and a receipt of the total donation. We tell them that in the spreadsheet the participants will only be identified by the last 5 characters of their Prolific IDs, to prevent them from taking into account others' perception of them.

After a participant completed the experiment, we sent them the monetary bonus they specified in the Bonus & Donation stage through Prolific. Donations from the participants totaled

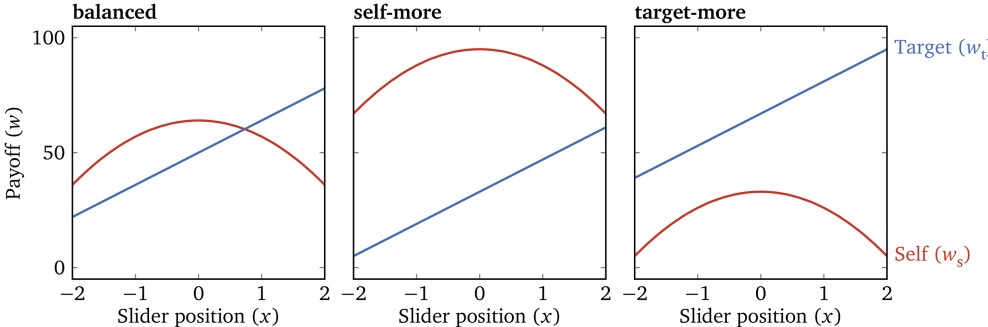

**Fig 8. Payoff functions of the three quadratic Lambda Sliders in Experiment 3.** The ranges on the $x$ axes reflect the ranges of the sliders.

$65.01, and we donated this amount to the Maui Strong Fund and sent the participants a message through Prolific with links to the spreadsheet and receipt as we promised them.

## Results

**Robustness.** We first fit a 6-variate normal distribution (2 measurements $\times$ 3 sliders for each participant–target combination) to the data from the Slide stage to examine the within-slider and between-slider correlations (see S4 Appendix for details). The within-slider correlations are ("b" for balanced, "s" for self-more, "t" for target-more) $\rho_b = 0.853\ (0.825, 0.879)$, $\rho_s = 0.889\ (0.863, 0.911)$, and $\rho_t = 0.868\ (0.841, 0.890)$, confirming that the Lambda Slider has high test–retest reliability for a variety of configurations, even though its scale is smaller in this experiment than previous ones ($a = 7$ vs. $a = 11.25$; Figs 2B, 6 and 8). The Bayes factor between the full model and an alternative model where $\rho_b = \rho_s = \rho_t$ is roughly 1.6, indicating inconclusive evidence about whether the test–retest reliabilities of the three sliders are meaningfully different. The between-slider correlations are $\rho_{bs} = 0.802\ (0.768, 0.831)$, $\rho_{bt} = 0.784\ (0.751, 0.814)$, and $\rho_{st} = 0.681\ (0.629, 0.728)$, indicating that measurements of $\lambda$ are relatively robust to different shift parameters $b_s$ and $b_t$.

We examined the relationship between $\lambda$ and the social distance ranking (excluding the Maui wildfire victim) with the same model as in Experiment 2 (see S4 Appendix), and the mean slope is $b_3 = -0.60\ (-0.73, -0.47)$. (Since there are only 5 targets here, the value of this slope is roughly comparable to the previous experiments after being divided by 2.) Given the larger sample size compared to Experiments 1 and 2, we also conducted an exploratory analysis of the effect of sex on $\lambda$ and the interaction between sex and social distance ranking, and found no evidence toward the existence or nonexistence of these two effects, meaning that the sample size is still not large enough to reach a conclusion (see S4 Appendix).

**External validity.** To examine the relationship between measurements on the Lambda Slider and real-world altruistic behavior, we fit a 3-variate normal distribution to the participants' measured $\lambda$s toward the Maui wildfires victim and their actual donations to the Maui Strong Fund (see S4 Appendix). We fit the model separately for the three different sliders. For the balanced slider, the mean and standard deviation of $\lambda$ are $\mu_\lambda = 1.15$ and $\sigma_\lambda = 1.39$, and the mean and standard deviation of participants' donations are $\mu_d = 0.34$ and $\sigma_d = 1.60$. (Here $\sigma_d$ is larger than the maximum possible standard deviation of data points bounded between 0 and 2, because many (58%) donation amounts are either 0 or 2 and they are treated as censored data.) The correlation between $\lambda$ and the donation is $\rho_{\lambda d} = 0.448\ (0.231, 0.618)$, and the Bayes factor between the full model and a null model where $\rho_{\lambda d} = 0$ is BF = 640, indicating extreme evidence that the measured $\lambda$ and the donation are positively correlated and that the Lambda Slider has good external validity. This correlation is close to the correlation of 0.42 between the SVO Slider Measure (in terms of SVO°) and a standard dictator game [31], despite the Lambda Slider only depending on 1 response instead of 6. For the self-more slider, $\mu_\lambda = 1.93$, $\sigma_\lambda = 1.77$, $\mu_d = 0.32$, $\sigma_d = 1.60$, $\rho_{\lambda d} = 0.453\ (0.245, 0.623)$, BF = 491. For the target-more slider, $\mu_\lambda = 0.93$, $\sigma_\lambda = 1.26$, $\mu_d = 0.35$, $\sigma_d = 1.58$, $\rho_{\lambda d} = 0.349\ (0.139, 0.529)$, $p_d = 99.95\%$, BF = 33.3.

**Inequity aversion.** If participants are inequity-averse, i.e., they have a non-zero $\kappa$, the slider position they choose on average would be highest on the self-more slider, lowest on the target-more slider, and in-between on the balanced slider. In the fitted 6-variate normal distribution described above, we have $\mu_s = 1.04\ (0.85, 1.23)$, $\mu_t = 0.08\ (-0.02, 0.18)$, and $\mu_b = 0.22\ (0.10, 0.33)$, suggesting that participants are indeed inequity-averse to some extent.

We fit a hierarchical model to jointly estimate $\lambda$ and $\kappa$ for each participant–target combination. We assume that each participant has a fixed $\kappa$, but their $\lambda$ varies across targets. We

restrict the range of $\kappa$ to $[0, 0.95]$ because the model becomes unstable when $\kappa$ gets too close to 1 (see S4 Appendix for details and other assumptions).

Fig 9A plots the estimates of $\kappa$ for each participant, which span a wide range. Figs 9B–D plot raw responses of three participants with high, medium and low estimates of $\kappa$. We see that the higher $\kappa$ is, the more slider positions are influenced by the relative offsets of the sliders. The vertical distance between a cross (predicted response given inferred $\lambda$ and $\kappa$) and a horizontal line (inferred $\lambda$) reflects how much a "naïve" measurement of $\lambda$ based on the slider position alone is biased from a more sophisticated measurement that takes into account inequity aversion.

To examine the external validity of $\kappa$, we look at the relationship between a participant's estimated $\kappa$ and how far the participant's donation $d$ is from the equal-payoff point: $|d-1|$. The two variables are negatively correlated ($\rho = -0.343$ ($-0.527, -0.129$), $p_\mathrm{d} = 99.92\%$), indicating that participants with a higher $\kappa$ are more likely to choose equal payoffs between themselves and another person in real-world decisions.

These data suggest that there is considerable variation among participants in terms of the degree of inequity aversion and, although a single response on the Lambda Slider is highly

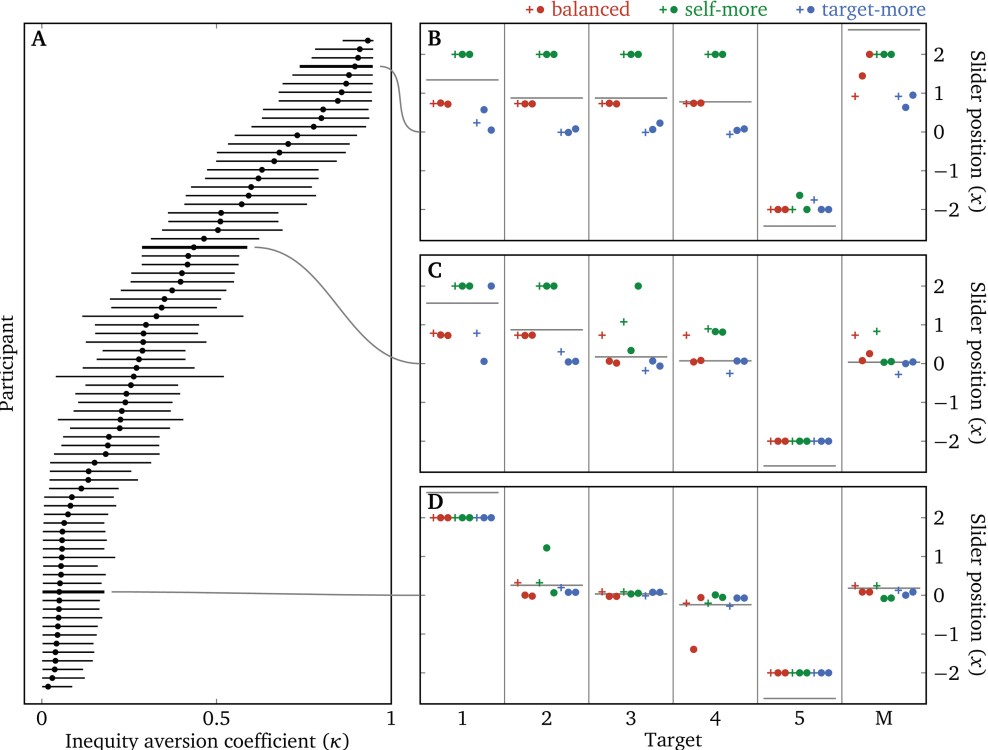

**Fig 9. Inequity aversion in Experiment 3. (A)** Posterior distributions of $\kappa$ for each participant sorted by posterior median. The dots indicate the posterior medians and the lines indicate the 95% credible intervals. Three participants are highlighted, whose raw responses are plotted in **(B)–(D)**. Targets 1–5 are the targets listed by the participants in the List stage, in increasing order of social distance. The target "M" is the Maui wildfires victim. For each participant–target combination, the horizontal line represents the posterior median of $\lambda$. For each participant–target–slider combination, the cross represents the predicted utility-maximizing response given the posterior medians of $\lambda$ and $\kappa$, while the two dots are the actual responses.

correlated with a participant's true $\lambda$, it may be biased toward the equal-payoff point, especially for participants with high degrees of inequity aversion. In many research programs such biases do not affect the validity of the conclusions, but if and when such biases are a concern, we recommend that researchers jointly estimate $\lambda$ and $\kappa$ using multiple Lambda Sliders. We also recommend fitting a complete model like we did for the benefits of having uncertainty estimates and easy integration of prior and global information. But a quick point estimate of $\lambda$ and $\kappa$ is also possible by having one measurement $x_1$ where $w_s > w_t$ and another measurement $x_2$ where $w_s < w_t$ on two sliders with different relative offsets, and then solving

$$x_1 = \frac{\lambda + \kappa}{1 - \kappa}, \tag{27}$$

$$x_2 = \frac{\lambda - \kappa}{1 + \kappa} \tag{28}$$

for $\lambda$ and $\kappa$ by virtue of Eq. (26):

$$\lambda = \frac{x_1 + x_2 + 2x_1 x_2}{2 + x_1 + x_2}, \tag{29}$$

$$\kappa = \frac{x_1 - x_2}{2 + x_1 + x_2}. \tag{30}$$

There is a solution for $\kappa \in [0, 1)$ as long as $x_1 \geq x_2 > -1$ or $x_1 \leq x_2 < -1$. Otherwise, we cannot obtain an estimate of $\kappa$ in this way but can use the average of $x_1$ and $x_2$ as a point estimate of $\lambda$.

Now we can have a refined understanding of the tradeoff between accuracy and efficiency discussed in the Introduction. Accuracy entails both unbiasedness and reliability. There is a straightforward tradeoff between reliability and efficiency for any measure; the more administrations of a measure are averaged to get a single measurement, the more reliable the measurement will be. On the other hand, biases are trickier to deal with, and none of the correlation metrics we reported in the experiments really deals with biases. In the context of measuring $\lambda$, biases are prominently introduced in two ways: (a) through discreteness in the underlying measure, such as the measures based on binary allocation tasks; and (b) through the failure of accounting for inequity aversion. The Lambda Slider, unlike most other measures of $\lambda$, is free of the first kind of biases. The second kind of biases can be mitigated by administering multiple Lambda Sliders with different relative offsets of the payoff functions for each participant–target combination and jointly estimating $\lambda$ and $\kappa$, assuming that they are stable across the multiple measurements. Of course, one has to sacrifice some efficiency for this joint estimation. In general, the more prior information one has about $\lambda$ and/or $\kappa$, the less efficiency one has to sacrifice to achieve the same level of accuracy.

## Discussion

We have developed the Lambda Slider, an accurate and efficient measure of $\lambda$ that is theoretically rigorous. We have shown that the Lambda Slider has high reliability, convergent validity, and external validity for real-world decisions. We have also demonstrated how multiple Lambda Sliders can be used to correct the biases in the measurements of $\lambda$ caused by inequity aversion.

The Lambda Slider can be straightforwardly implemented using any dynamic graphical user interface. To make it easier for other researchers to use the Lambda Slider, we have created a standalone version of the quadratic Lambda Slider with the same payoff functions as in

Experiment 1 (https://experiments.evullab.org/lambda-slider/). It can be directly embedded into web-based survey platforms such as Qualtrics; instructions can be found in the README file at https://doi.org/10.5281/zenodo.14563524.

We did not record how exactly participants moved the slider in each trial. Future studies can record this, which might provide insight into participants' mental processes when making a choice on the slider. For instance, if participants move the slider from its initial position directly toward its final position without moving back and forth, it suggests that they are following the gradient of the utility function and finding the global maximum in an efficient way. Participants could also restrict their search to a small range after being familiarized with the Lambda Slider, which would be reflected by a jump from the initial position of the slider followed by small local movements. Such movement data could also be used to assess the degree to which participants explored the full range of payoffs to make decisions, and whether additional instructions to explore the slider affect its psychometric properties.

Although the Lambda Slider is efficient and has good psychometric properties, it may not be the best measure to use under some conditions. The nonlinear payoff structures may be difficult for people to quickly familiarize themselves with. It is also inapplicable to projects relying on paper-based measures. Under these circumstances, it may be preferable to use another measure such as the SVO Slider Measure [18] or the Welfare Trade-Off Task [14,15]. The SVO Slider Measure may also better align with personality scale measures designed to assess the same four social strategies that serve as endpoints for the SVO items.

One potential future direction is to use the Lambda Slider to study social perception. People not only make social decisions based on their $\lambda$s toward other people, but can represent, infer, and predict others' $\lambda$s toward themselves or someone else and react accordingly (e.g., [9,32–37]). Because of the drawbacks of the existing measures of $\lambda$ based on binary allocation tasks, the processes of (a) conveying another person's $\lambda$ to the participant, and (b) measuring the participant's *prediction* of another person's $\lambda$, have had a relatively low ceiling on the product of accuracy and efficiency, limiting the study of the dynamics of such inference and prediction over time or space. The Lambda Slider can potentially be used to make these processes more accurate and/or efficient. Using the Lambda Slider to measure participants' predictions of another person's $\lambda$ seems straightforward—Alice could imagine that she adopts Bob's $\lambda$ and makes decisions on the Lambda Slider in the same way she makes her own decisions. Participants should also be able to infer others' $\lambda$s from observations of Lambda Slider choices, so long as the observing participants have a good understanding of the underlying payoff functions. This understanding could potentially be achieved by allowing the participant to manipulate the slider, or by depicting the relationship between the payoff functions using a 2D curve, as in Fig 1D. The validity and reliability of the (1D or 2D) Lambda Slider for either of these purposes need to be established by further research.

## Supporting information

**S1 Appendix Formal derivation of the Lambda Slider.**
(PDF)

**S2 Appendix Circle Test and circular Lambda Slider.**
(PDF)

**S3 Appendix Payoff functions in Experiments 2 and 3.**
(PDF)

**S4 Appendix Model specifications.**
(PDF)

## Author contributions

**Conceptualization:** Wenhao Qi, Edward Vul, Lindsey J. Powell.

**Data curation:** Wenhao Qi.

**Formal analysis:** Wenhao Qi.

**Funding acquisition:** Edward Vul.

**Investigation:** Wenhao Qi.

**Methodology:** Wenhao Qi, Edward Vul, Lindsey J. Powell.

**Project administration:** Edward Vul, Lindsey J. Powell.

**Software:** Wenhao Qi.

**Supervision:** Edward Vul, Lindsey J. Powell.

**Validation:** Wenhao Qi, Lindsey J. Powell.

**Visualization:** Wenhao Qi.

**Writing – original draft:** Wenhao Qi, Edward Vul, Lindsey J. Powell.

**Writing – review & editing:** Wenhao Qi, Lindsey J. Powell.

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
