## [Decision Letter · Decision Letter 0]

12 Nov 2024

PONE-D-24-43900An accurate and efficient measure of welfare tradeoff ratiosPLOS ONE

Dear Dr. Qi,

Thank you for submitting your manuscript to PLOS ONE. After careful consideration, we feel that it has merit but does not fully meet PLOS ONE’s publication criteria as it currently stands. Therefore, we invite you to submit a revised version of the manuscript that addresses the points raised during the review process.

I have received referee reports from two experts in this area of research and they have provided excellent feedback. Reviewer 1 suggests a major revision. Reviewer 2 suggests a minor revision. I would like to echo Reviewer 2 in that I think this paper has great potential to make a nice contribution to the literature but would benefit from addressing the concerns in the referee reports. I encourage the authors to address each of the reviewers suggestions and comments. Additionally, I have some suggestions that I think may help as well. Below I highlight a few main areas I encourage the authors to address in the revision:

Sample size: Reviewer 1 discussed a concern that both experiment 1 and 2 have a very small sample sizes. The authors should address those comments. Additionally, more information is needed about how the sample size for the first experiments were determined. While the authors are correct that Bayesian analysis can handle small samples, this means that the prior will have a stronger influence on the results. It would be helpful for the authors to discuss their prior and how it was determined. Was the prior chosen as weakly informative or based on prior SVO data? More discussion is needed. It may be helpful to run a post-hoc power analysis to see the degree in which the first two experiments may be underpowered. Additionally, were the results from experiments 1 and 2 used for the power analysis for the preregistration for experiment 3? More details would be helpful.

Experimental Instructions: I tried out the lamda task experiment using the link provided by the authors in the text. One issue I saw is that in the non-zero sum distribution questions, the only way a subject will know the full range of choices is if they move the slider in both directions. It is not clear in the directions that subjects should try to move the slider in both directions for the full range to check their choices. In the software is there any recording whether subjects moved the slider the full range to see all the possible distribution options? If so, then this data may be worth checking to see if this introduced any bias in the elicitation method. Additionally, it may be worth discussing this as a possible drawback as the elicitation method may introduce some error in elicitation if some subjects do not move the slider to check the full range of choices.

Comparison to SVO: Both reviewers highlight that the paper would benefit from more discussion about how the lamda slider compares and contrasts to SVO.

Data Analysis: Reviewer 2 has some great suggestions about providing more data analysis and suggests adding some descriptive statistics for each experiment. There were multiple measures in the experiment and provided descriptive statistics for each measure would help.

We look forward to receiving your revised manuscript.

Kind regards,

Garret Ridinger

Academic Editor

PLOS ONE

Journal Requirements:

Reviewers' comments:

Reviewer's Responses to Questions

**Comments to the Author**

1. Is the manuscript technically sound, and do the data support the conclusions?

Reviewer #1: Partly

Reviewer #2: Yes

2. Has the statistical analysis been performed appropriately and rigorously? 

Reviewer #1: Yes

Reviewer #2: Yes

3. Have the authors made all data underlying the findings in their manuscript fully available?

Reviewer #1: Yes

Reviewer #2: Yes

4. Is the manuscript presented in an intelligible fashion and written in standard English?

Reviewer #1: Yes

Reviewer #2: Yes

5. Review Comments to the Author

Reviewer #1: The authors introduce the Lambda Slider, a novel measure of social preferences, based on a welfare tradeoff ratio (WTR). The major benefit of this measure is that it uses a single continuous slider to give a more precise measure of the concern for others, which would require participants to make multiple binary allocations in previous methods such as the SVO.

Experiment 1 tests the reliability of the Lambda Slider and SVO by eliciting two measurements for each participant. The authors find that correlations between the two measurements are high for both the Lambda Slider and SVO. The two measures are also positively correlated (convergent validity). Experiment 2 provides a robustness check and shows that participants are not simply moving the position of the Lambda Slider without any regard for the corresponding allocation values. Experiment 3 tests external validity by asking participants to make a donation. The authors show that the measure is positively correlated with donations. Experiment 3 also cautions that the measure may be biased by inequality aversion.

The paper presents a measure that is useful to researchers studying social preferences with nice benefits over existing methods. I appreciate the robustness check conducted in Experiment 2 as well as the test of external validity using an incentivized charitable donation in Experiment 3.

However, my main concern is with the sample sizes of Experiment 1 and 2. In Exp 1 only 40 participants were recruited with 30 remaining after excluding those who failed attention checks. While in Exp 2 only 20 were recruited with only 16 passing the checks. Given that the reliability of the Lambda Slider is a major selling point and a central part of the paper, I am surprised that so few observations were collected. While for Experiment 3 (which I understand was the only preregistered experiment), 90 observations were collected.

When reading the Abstract and Introduction, the contribution of this particular paper was not fully clear to me. This may be personal preference but I found the Introduction to be quite long and took quite a while before any discussion of what is done in this paper and how it differs from previous work. I feel that a discussion of the SVO should come much earlier in the Introduction (and should also be mentioned in the Abstract) given that it is what comes to mind for most people for measuring social preferences. Alongside this, I think the authors should emphasise more what sets the lambda slider apart from the SVO and especially the advantages over the SVO.

Related to the previous point, when the authors discuss the strengths of the lambda slider it would be helpful to define the criteria discussed throughout the paper in the Introduction, e.g., terms such as reliability, efficiency, accuracy, and sensitivity. This would help to convince the reader early on of the advantages of this method over previous methods.

Reviewer #2: The manuscript introduces a new measure of welfare-tradeoff ratios called the lambda slider. In three experiments, the authors show that the measure has high internal, convergent and external validity, and high reliability. They also find that inequity aversion can bias welfare-tradeoff ratios measurements, but that the new measure can be used to control for inequity aversion.

I thought this was an outstanding paper. In addition to having practical implications for measurement, it is also an elegant and insightful piece of theoretical work. The section on inequity aversion in particular provides very valuable new insights. The logic of the new measure is clearly explained, and the paper is in general very well written. Also the Bayesian analysis and open science practices are exemplary.

Most of the comments below are motivated by my own curiosity, and I don’t think any of them are crucial to the main conclusions of the paper.

-It might be useful to the reader if the relationship between SVO and lambda was explicated more precisely. The manuscript says that the SVO measure is `similar’ to lambda, but this is ambiguous between `it measures the same thing in a slightly different way’ and `it measures a slightly different thing’.

-I think many readers will be curious about the descriptive statistics. For example, the authors report the average wtrs for each slider in experiment 3, but this could also be reported for the other experiments (as well as the standard deviations). A similar point holds for the descriptive statistics of Dictator Game giving in experiment 3, which I didn’t find.

-One way to measure the reliability of the measure is to assess test-retest reliability, either within or between sliders, as the authors do. But another way is to test to what extent different sliders yield the same average wtr. Suppose for example that two sliders A and B have a .95 between-slider reliability, but mean wtr is .2 when measured by A, and .6 when measured by B. Then there is a sense in which the slider method does not reliably tap into `the true wtr’ of a participant. This might of course not be a concern for certain research purposes (for example for experiments only interested in explaining variance in wtr), but could be relevant for others. This issue is one reason why it would be useful to report descriptive statistics for each slider in experiment 2.

-On a related note, in experiment 3 the measured wtr is clearly not slider-invariant, as helpfully reported in the text. As the authors convincingly argue, this non-invariance can be explained as a result of inequity aversion. So, if we find that slider-invariance also fails to hold in experiment 2, this might simply mean that inequity aversion is also implicated in that experiment.

-To be clear, I think that the issue described above points to a *strength* of the present method. The flexibility in the parameterization of the measure allows researchers to explore theoretical issues which are also a concern for other measures but difficult to investigate with these measures.

-One question raised by Experiment 3 is to what extent we should in general expect inequity aversion to bias our estimates of WTR. The comparison between the three sliders suggests that this bias can be subtantial (the average measured wtrs range from .08 to 1.04). But this comparison doesn’t yet tell us whether inequity aversion can bias our measure of relative WTRs. For example one can imagine that when measuring WTR in the `naïve way’ we estimate that participant A’s wtr is greater than participant B’s, but that when controlling for inequity aversion (as the authors do) we find that B’s wtr is higher. So it might be instructive to run a `naïve’ analysis of experiment 3 (where one just infers wtr instead of wtr and inequity aversion jointly), to see how much the results from that analysis might diverge from the more mature one reported here.

-In experiment 2, some readers might be interested in the between-slider correlations.

-One possible direction for future work that could be mentioned is comparison of the lambda slider with the Welfare-Tradeoff Task.

-I was impressed by the clarity of the way the slider measure is motivated by imagining adding more and more options to an n-tuple dominance task. Maybe a complementary way of conveying intuition is to remark that each point of the slider corresponds to one particular exchange rate between the participant and the target’s welfare. I.e. what makes the measure efficient is that the participant can vary the exchange rate continuously until she finds the one that corresponds to her preferred one. (The authors should feel free to ignore this remark if this makes that section of the paper too long).

-when re-deriving the equation for k (between lines 569 and 570) I obtained a slightly different result, k=(x1-x2)/(2+x1+x2). It is very possible the mistake is on my end but it might be worth checking for a possible sign error.

-(very minor) I was curious about the pattern of correlations between WTR, inequity aversion, and DG giving. For example if we predict DG giving from WTR and inequity aversion jointly, does one variable come out as a better predictor? But maybe some of these questions are difficult to address rigorously, because experiments are not powered with these analyses in mind and the inequity aversion of a participant is measured from more data points and so potentially more precise.

- (very minor) To help readers keep track of when x (position on a slider) directly maps to lambda, maybe the authors could say that it does so in all experiments unless indicated otherwise.

6. PLOS authors have the option to publish the peer review history of their article (what does this mean?). If published, this will include your full peer review and any attached files.

Reviewer #1: No

Reviewer #2: No

---

## [Author Response · Author response to Decision Letter 1]

31 Dec 2024

Please see the attached response letter.

---

## [Decision Letter · Decision Letter 1]

7 Feb 2025

PONE-D-24-43900R1An accurate and efficient measure of welfare tradeoff ratiosPLOS ONE

Dear Dr. Qi,

Thank you for submitting your manuscript to PLOS ONE. After careful consideration, we feel that it has merit but does not fully meet PLOS ONE’s publication criteria as it currently stands. Therefore, we invite you to submit a revised version of the manuscript that addresses the points raised during the review process.

We look forward to receiving your revised manuscript.

Kind regards,

Garret Ridinger

Academic Editor

PLOS ONE

Journal Requirements:

Reviewers' comments:

Reviewer's Responses to Questions

**Comments to the Author**

1. If the authors have adequately addressed your comments raised in a previous round of review and you feel that this manuscript is now acceptable for publication, you may indicate that here to bypass the “Comments to the Author” section, enter your conflict of interest statement in the “Confidential to Editor” section, and submit your "Accept" recommendation.

Reviewer #1: All comments have been addressed

Reviewer #2: (No Response)

2. Is the manuscript technically sound, and do the data support the conclusions?

Reviewer #1: Yes

Reviewer #2: Yes

3. Has the statistical analysis been performed appropriately and rigorously? 

Reviewer #1: Yes

Reviewer #2: Yes

4. Have the authors made all data underlying the findings in their manuscript fully available?

Reviewer #1: Yes

Reviewer #2: Yes

5. Is the manuscript presented in an intelligible fashion and written in standard English?

Reviewer #1: Yes

Reviewer #2: Yes

6. Review Comments to the Author

Reviewer #1: I am satisfied that the authors have done a good job in addressing my comments and other comments raised.

Reviewer #2: The authors have addressed most of my initial remarks. Below are a few more comments.

-On p.18 of the revised manuscript (tracked changes version), line 414, I was very confused by the sentence:

`Under H_lambda, the mean and standard deviation of X_base are mu=.58 and sd=.27 (Xpos and Xneg have means defined by eq 9 […])’.

In my understanding of the setup, X_base is a purely empirical measure (it is the position of the participant’s slider for the base slider), so I did not understand why this was contextualized by ‘under H_lambda’. Furthermore, equation 9 describes the predictions of the theory; and saying that we can get the empirical means for Xpos and Xneg in this way seems to already pre-suppose that the theory is right, but checking this is exactly why we are examining the empirical data in the first place. It is instead highly possible that Xpos and Xneg don’t have empirical means at exactly the value expected by the theory (even just for simple noise reasons).

I also think the authors should plot these empirical means of lambda for each slider (e.g. as bar graphs) along with the model predictions (for H_lambda and H_X). The Figure with bivariate distributions is great but demands a lot of attention from the reader, the better fit of the H_lambda hypothesis doesn’t jump out at the reader immediately.

-On p.24 of the revised manuscript (tracked changes version), the sentence starting with ‘For the balanced slider […]’ (line 561) is potentially confusing because it mentions the descriptive results for the balanced slider in the same sentence as the descriptive results for the dictator game, in a way that might implicitly suggest that the dictator game is a ‘balanced’ dictator game and that we should expect the descriptive results for the ‘self-more’ and ‘other-more’ dictator games later (whereas in fact there is only one dictator game). Similarly this sentence reports the average inferred WTR for the balanced slider (which is very useful) but I wasn’t able to find the same information for the other sliders later.

-In response to my comment: `I was curious about the pattern of correlations between WTR, inequity aversion, and DG giving. For example if we predict DG giving from WTR and inequity aversion jointly, does one variable come out as a better predictor? But maybe some of these questions are difficult to address rigorously, because experiments are not powered with these analyses in mind and the inequity aversion of a participant is measured from more data points and so potentially more precise.’, the authors mention a possible approach based on model comparison where one would compare the fit of different regression models including either lambda or kappa as predictors (or both). But I had in mind a simpler and possibly more standard approach where one would simply run a regression of the type DG ~ lambda + kappa, and compare the magnitude of the standardized coefficients of both predictors.

7. PLOS authors have the option to publish the peer review history of their article (what does this mean?). If published, this will include your full peer review and any attached files.

Reviewer #1: No

Reviewer #2: No

---

## [Author Response · Author response to Decision Letter 2]

18 Mar 2025

Please see the attached response letter.

---

## [Editor Report · Decision Letter 2]

21 Mar 2025

An accurate and efficient measure of welfare tradeoff ratios

PONE-D-24-43900R2

Dear Dr. Qi,

Thank you for your revised manuscript. I am confident that you have addressed all outstanding comments from the reviewers and recommend acceptance of your paper. I think your paper will make an excellent contribution.

I am pleased to inform you that your manuscript has been judged scientifically suitable for publication and will be formally accepted for publication once it meets all outstanding technical requirements. 

Kind regards,

Garret Ridinger

Academic Editor

PLOS ONE

---

## [Editor Report · Acceptance letter]

PONE-D-24-43900R2

PLOS ONE

Dear Dr. Qi,

I'm pleased to inform you that your manuscript has been deemed suitable for publication in PLOS ONE. Congratulations! Your manuscript is now being handed over to our production team.

Kind regards,

on behalf of

Dr. Garret Ridinger

Academic Editor

PLOS ONE